# Effect of Village Informal Institutions and Cadre-Mass Relationship for Farmers’ Participation in Rural Residential Environment Governance in China

**DOI:** 10.3390/ijerph20010003

**Published:** 2022-12-20

**Authors:** Diandian Hao, Ziyi Yan, Yanan Wang, Bowen Wang

**Affiliations:** 1College of Economics and Management, Northwest A&F University, Xianyang 712100, China; 2College of Foreign Languages, Inner Mongolia University, Hohhot 010021, China

**Keywords:** rural residential environment governance, informal institutions, village cadres, mobilized governance

## Abstract

Rural residential environment governance (RRE), as the first tough battle of China’s rural revitalization strategy, relies on farmers’ participation since farmers are the main laborers, builders, and administrators in environmental governance. However, lackluster farmers’ enthusiasm and initiative have hindered RRE initiatives, prompting this paper. Based on the survey data of 1804 farmers in China, this paper, from the perspective of mobilization governance, empirically analyzes the impact of informal village institutions, the cadre-mass relationship, and their interaction on farmers’ participation in RRE governance through the entropy method, ols, and quantile regression model. The empirical results indicate that informal institutions promoted farmers’ participation through three mechanisms, with disciplinary supervision impact being the most significant and value-oriented next, but transmission internalization doesn’t work as well as it should. Meanwhile, for farmers with varying degrees of participation, there is a general difference in the governance effect of informal institutions. Furthermore, the close cadre-mass relationship significantly strengthened disciplinary supervision and transmission internalization effects to mobilize farmers’ participation. Therefore, the village committee should implement diverse informal institutions based on the actual situation of their village. Moreover, it is necessary to shape a close cadre-mass relationship to improve the accuracy of institutions’ implementation.

## 1. Introduction

In developing countries, the problem of unbalanced urban and rural development and insufficient rural development is prominent. The most intuitive expression is the obvious gaps between urban and rural residential environments [1,2]. The UN Conference on Housing and Sustainable Urban Development (Habitat III) reaffirmed the commitment of governments to a sustainable human residential environment [3]. How to achieve the sustainable development of human settlements at different levels (cities, towns, and countries) is still a major issue facing the world [4]. As the largest developing country in the world, China has a large population pressure, insufficient resources, and limited environmental carrying capacity [5,6]. Due to the inherent imperfection of the ecological environment and the rigid constraints of economic growth, the contradiction between economic development and environmental protection is more prominent [6]. Especially in China’s pre-development period, limited by the urban-rural dual institutions, rural environmental governance has lagged far behind the city [7,8]. Against this backdrop, the Chinese government has put rural environmental governance on its agenda and introduced policies, and called on governments at all levels and grassroots organizations to make the issue a priority [9,10,11]. Recently, China’s State Council has promulgated successive policy measures such as *the Three-Year Action Plan for Rural residential environment Governance (2018–2020)* and *the Five-Year Action Plan for rural residential environment governance and Upgrading (2021–2025)*. As reflected in the transition from “three years of action” to “five years of improvement” in the document, China’s RRE revitalization has entered a new phase based on strengthening comprehensive governance effectiveness. [11].

Rural residential environment governance has received unprecedented attention from the Chinese government [9,10]. With joint efforts, the rural living environment has improved [12]. However, the Chinese government ‘s traditional ‘top-down‘ governance model has encountered bottlenecks in advancing RRE governance [11,13]. The RRE governance process encountered resource utilization is inefficient, falling into the dilemma of “high input, low efficiency.” The main reasons are farmers excessive dependence on government funds and support, and enthusiasm and initiative are lacking [14,15,16]. However, farmers are the primary stakeholders in RRE, assuming the roles of maintainer, super-visor, and beneficiary [17,18]. Without the effective participation of farmers, the effectiveness and sustainability of RRE governance are weakened [16,19]. Thus, mobilizing the active participation of farmers is crucial for building and maintaining livable villages and meeting long-term development goals.

It is difficult to avoid farmers’ free-riding behavior due to the nature of public goods nature, the diversity of interests, and the complexity of the environmental problem. The new institution of the economic theory states that there are two types of institutions: formal and informal. North holds the point that the informal institution is an agreed-upon and standard code of conduct that people gradually form in the long-term social interaction process and are recognized by society [20]. Existing research has demonstrated that in addition to formal institutions, farmers’ behavior is also influenced by informal institutions [10,21,22]. As Dlangalala et al. found, informal rules positively impacted smallholder awareness of water management issues related to collective irrigation in South Africa [23]. Moreover, Mastewal et al. found that informal institutions were critical for increasing Ugandan farmers’ investment in sustainable crop intensification. Yu Cao et al. reported that informal institutions positively impacted Chinese farmers’ use of clean energy [24,25]. In summary, the importance of informal institutions on farmers’ behavior. In China, villager autonomy is the fundamental political system to maintain rural governance, and committees elected by villagers are responsible for managing rural public affairs [26]. The village committee forms a set of village regulations based on local customs and farmers’ consensus opinions [22]. These regulations are the important institutional basis of villager autonomy, which enables members to carry out self-management, self-education, and self-restraint [20]. Village regulation is the most significant and extensive part of the Chinese rural informal institutions. Village regulation is the most significant and extensive part of the Chinese rural informal institutions [21]. Thus, what is the effect of the informal institutions represented by village regulations on RRE? The question deserves further consideration.

Whether institutions can be recognized, understood, and strictly implemented in the RRE process requires not only government intervention but also active docking at the micro level, such as rural grassroots organizations [27]. In China, village committees are semi-governmental, with the government “steering” and village committees “rowing,” and village committees playing a critical role in facilitating and coordinating RRE governance. Village committees, which are staffed by village leaders who are elected by the villagers, have a natural connection with local farmers and their customs and traditions [28]. Thus, village cadres perform dual roles of “government agents” and “farmers’ stewards” by assisting the government in performing “official duties” while managing “village affairs” for villagers [29]. As far as RRE governance is concerned, the positive interactive relationship between village cadres and villagers is embodied in the exchange and sharing of resources for the unified goal. Eventually, they form a long-term and stable relationship of mutual dependence and cooperation [25]. The quality of the relationship between cadres and farmers will further affect the mobilization effect of farmers [29]. Studies indicate that the close cadre-mass relationship impacted agricultural waste recycling, infrastructure management, and land expropriation compensation efforts [22,25,30] and maintains village discipline while standardizing farmers’ cooperation and collective action.

To the best of our knowledge, the contribution of this paper to the existing literature is as follows. On the one hand, concerning the research subject, as China’s RRE governance advances, differences in farmers’ behavior are no longer simply a matter of whether or not to participate but more of the degree of participation. However, existing research on the degree of farmers’ participation has mostly focused on a particular aspect of RRE governance or is based on farmers’ evaluation of their own participation. There is a lack of multidimensional measurement of farmers’ actual participation in RRE governance. Thus, this study explores and consummates the evaluation index system to measure the degree of farmers’ participation in RRE governance and enrichment of the existing literature. On the other hand, concerning the research perspective, under the villager autonomy system of rural society in China, environmental village regulations and village cadres are important factors in mobilizing farmers to participate in the governance of public affairs. Institutions are undifferentiated for any individual. However, in the “pattern of difference sequence“ centered on village cadres, village cadres cannot be undifferentiated for all villagers. What is the impact of the cadre-mass relationship on the implementation of the informal village institutions, and whether there is an interaction between the two is worth exploring. However, most of the previous studies analyzed the rural informal institutions or CMR from a single perspective, rarely into a unified framework for specialized exploration. The answers to the above questions have important practical significance to comprehensively improve the quality of the rural living environment, narrow the gap between urban and rural areas and build a moderately prosperous society in an all-around way.

In view of this research gap, based on the survey data of 1804 farmers in Shaanxi Province, China, this paper, from the perspective of mobilization governance, empirically analyzes the impact of informal village institutions, the cadre-mass relationship and their interaction on farmers’ participation in RRE governance through the entropy method, ols, and quantile regression model. Specifically, this research addresses the following three main targets: (1) How and to what extent do informal institutions, and cadre-mass relationships affect the degree of farmers’ participation in RRE governance? (2) What is the interactive influence of informal institutions and cadre-mass relationships on farmers’ participation in RRE governance? (3) What are the changing trends in the impact of informal institutions and cadre-mass relationships on farmers’ participation in RRE governance?

## 2. Theoretical Framework and Research Hypothesis

### 2.1. Theoretical Framework

As a kind of social governance system with strong Chinese characteristics, mobilized governance has a deep historical origin [31]. As early as the founding of the Communist Party of China, the social mobilization mechanism was established on the basis of the mass line of “reasoning by truth, touching with heart.” Since then, it has been developed into a “mobilized governance” that is effective and different from “hierarchical governance” and “market-oriented operations.” Through various forms of propaganda, mobilization, and organizational work, mobilized governance enables the general public to develop certain values, living habits, attitudes, etc., so that it produces a continuous governance model that obeys instructions or other expected behaviors [32]. At the current stage, it is difficult to achieve effective environmental governance by relying solely on the external forces of the government, and mobilizing the participation of farmers, and various forces of rural society, has become an important way of RRE management [33]. Based on this, the paper constructs an analytical framework for the effectiveness of mobilizing farmers to participate in RRE. The article’s theory analysis is shown in Figure 1 in detail

### 2.2. Research Hypothesis

New Institutional Economics holds that any economic activity is a rational activity that maximizes utility under certain institutional constraints. In other terms, it is the individual’s socioeconomic activities that are strongly influenced by institutional factors. According to existing research, informal institutions can be divided into the following three dimensions according to their mechanisms of action [22,34,35]. The first regulation type pertains to value-oriented regulations that involve the establishment of model titles, model household examples, and rewards for good behavior. The second type involves disciplinary supervision regulations that rely on public blame, loss of reputation, and other punitive measures for unacceptable behavior. The third type pertains to the transmission internalization mechanism, whereby village rules and regulations are readily recognized and used to transmit values that can be easily internalized as a means of transforming undesirable behaviors into desirable behaviors. Based on the premise that the informal institutions represented by village regulations will affect the choice logic of farmers in the RRE governance process, we propose the following hypothesis:

**H1.** *Informal institutions can significantly increase the degree of farmers’ participation in RRE governance*.

The embedding of social capital within environmental governance processes has become an important tool for addressing the plight of governance [36]. In Chinese agrarian society, kinship and geographical factors have supported the development of a solid social network based on “key minority” societal members who can influence farmers’ behaviors [37]. The skills, prestige, and resource advantages of village cadres have enabled them to serve as “key minority” members [35], whereby the close relationship between cadres and the masses permits cadres to effectively influence farmers’ environmental practices. In addition, a good CMR encourages the villagers to show a strong sense of trust. To some extent, farmers’ mastery of environmental governance regulations is enhanced by close cadre-mass communication, which also reduces concerns about false information, thereby increasing farmers’ trust in the institution. In turn, greater trust in village institutional rules increases farmers’ enthusiasm for environmental governance, which promotes greater compliance. In light of these findings, the following hypothesis is proposed in this research:

**H2.** *The close CMR can significantly promote farmers’ participation in RRE governance*.

According to the theory of Organizational Support, employees are better able to meet their social-emotional needs when they receive recognition, respect, and resources from their organizations, thereby contributing more to the interests of the organization [38]. The interaction between ordinary farmers and village cadres in projects involving rural public affairs governance, such as RRE governance, guarantees farmers’ democratic rights to participate in rural public affairs, increases the energy of the public discourse, and boosts farmers’ self-efficacy in rural public affairs. On the contrary, farmers may feel alienated when they perceive that no one is listening to their perspectives and concerns. According to the attitude generalization mechanism, such feelings can adversely affect the implementation of organizational decision-making, which, in turn, can reduce the effectiveness of informal institutions. Therefore, we propose the following hypothesis:

**H3.** *The close CMR strengthens the influence of informal institutions on farmers’ participation in RRE governance*.

## 3. Materials and Methods

### 3.1. Data Sources

The data in this paper were obtained from a field survey conducted by the research team from June to July 2022 in the Guanzhong region in the central part of Shaanxi Province. The Guanzhong region primarily consists of plain terrain that has highly dense agricultural populations and villages. This area has undergone rapid economic development since local natural conditions, regional characteristics, and customs have supported the relatively complete and successful implementation of numerous RRE projects there. The paper used stratified sampling, followed by equal probability random sampling, to determine the survey sample. First, among the 40 counties (districts) in the Guanzhong region, eight sample districts were selected based on regional economic conditions, in order: Linwei District, Fengxiang District, Dali County, Wugong County, Pucheng County, Fufeng County, Qishan County, and Chenchang District. Second, 3–4 townships were then selected according to the status of RRE governance in each county, and 3–5 natural villages were selected in each township. Finally, 15–20 farmers in each village were randomly selected for household surveys. In this survey, 1864 questionnaires were finally obtained. After eliminating outliers and missing values, 1803 valid samples were finally obtained, and the effective participation rate of the questionnaire was 96.67%.

### 3.2. Variable Selection and Descriptive Statistics

*Farmers’ participation degree.* According to *the Five-Year Action Plan for Rural Residential Environment Governance and Upgrading (2021–2025)*, we define RRE governance aims as follows: to promote the rural toilet revolution, promote rural garbage and sewage treatment, improve village appearance, and establish long-term management and protection mechanisms. RRE governance methods are designed to support the major role of farmers, increase policy support, and strengthen organizational security. By combining the actual overall situation observed in the sampled township, the following four dimensions were selected to estimate the degree of farmer participation: rural toilet revolution, domestic garbage, domestic sewage treatment, and village appearance. Meanwhile, this paper uses the entropy method to calculate the comprehensive score of farmers’ participation in RRE. Methods such as this are widely used because they can reduce bias resulting from subjective human factors in weight settings. The specific evaluation indexes and weights are shown in Table 1 (due to limited space, specific calculation steps can be obtained from the author).

*Informal institutions*. Village regulations are an informal institutional concept. Researchers have found that informal institutions have three basic elements: regulatory, normative, and cultural-cognitive. Its external forms and mechanisms are value-oriented, disciplinary supervision, and transmission internalization. Accordingly, this paper analyzes informal institutions in terms of these three indicators. Value-oriented regulation is measured by a questionnaire based on “Whether the village has honorary recognition institutions (health model households, clean farm households, etc.).” disciplinary supervision regulations are measured based on “Whether the village has punitive provisions for environmental damage (notice of notification, criticism, and education, etc.).” The transmission internalization mechanism is measured by “Whether the farmer agrees with relevant content of village regulations.”

*Cadre-mass relationship* (CMR). CMR was characterized by three variables: “Frequency of your interaction with village cadres,” “The degree of support from village cadres,” and “Your level of trust in village cadres.” It can be seen from Table 2 that factor analysis was used to measure the CMR. Cronbach’s α was 0.850, and the KMO value was 0.731. This indicates that the data have good validity and reliability, suitable for factor analysis.

*Control Variables.* In the correlation analysis, we take the factors that may affect farmers’ participation in RRE as the main control variables. As indicated in Table 3, these control variables include (1) Household characteristics (HC), including age, level of education, labor outflow, public status, and income; (2) Community features (CF), including farmers’ cognitive status and neighborhood relations; (3) Policy intervention (PI), including government subsidies; (4) Resource conditions (RC), including village distance to town and terrain. 

### 3.3. Methods

First, due to the type of variables studied and the size of the sample, an analysis of factors influencing farmers’ participation in RRE was conducted using an Ordinary Least Squares regression (OLS) regression model. The following model was established based on the abovementioned theoretical analysis according to Equation (1):(1)$$PD_{i}=\beta_{0}+\beta_{1}CMR_{i}+\beta_{2}II_{i}+\beta_{3}HC_{i}+\beta_{4}CF_{i}+\beta_{5}PI_{i}+\beta_{6}RC_{i}+\varepsilon_{i}$$
where $PD_{i}$ represents the participation degree of farmer i in RRE governance and $CMR_{i}$, $II_{i}$ refer to the core explanatory variables that this paper focuses on, namely, informal institutions and the *CMR*, respectively. Control variables include 𝐻𝐶_𝑖_, 𝐶*F*_𝑖_, *PI*_𝑖_, and 𝑅𝐶_𝑖_, which represent household characteristics, community features, government intervention, and resource conditions, respectively. 

Second, since OLS regression can only estimate the conditional expectation effect of explanatory variables on explained variables, the results are easily swayed by extreme values. Quantile regression can effectively overcome extreme values’ influence on estimation results while also providing all the information about the conditional distribution [39]. Specifically, quantile regression assumes that the quantile of the conditional distribution of the dependent variable is a linear function of the independent variable, so it is possible to construct a quantile regression of a dependent variable to demonstrate the effect of the independent variable on the dependent variable quantile. Quantile regression analysis was conducted in this work using Equation (2), as follows:(2)$$Quant_{\theta }(PD_{i}|X_{i})=\alpha_{\tau }X_{i}$$
where $X_{i}$ is the independent variable in Equation (1). $Quant_{\theta }(PD_{i}|X_{i})$ indicates the conditional quantile τ (0 < τ < 1) of $PD_{i}$ corresponding to the quantile θ given the independent variable X. $\alpha_{\tau }$ is a coefficient vector achieved by minimizing the absolute deviation as follows:(3)$$\overset{⌢}{\alpha_{\tau }}=\min_{\alpha_{\tau }}\left\{\sum_{i:PD_{i}\geq X_{\alpha }}^{n}\tau |PD_{i}-X_{i}\alpha_{\tau }|+\sum_{i:PD_{i}<X_{\alpha }}^{n}\left(1-\tau )|PD_{i}-X_{i}\alpha_{\tau }\right|\right\}$$

The third step is the analysis of the Interactive effects of informal institutions and the CMR. Based on Equation (1), this paper adds the interaction terms of the cadre-mass relationship and village rules.
(4)$$PD_{i}=\beta_{0}+\beta_{1}CMR_{i}+\beta_{2}II_{i}+\lambda CMR\times II_{i}+\beta_{3}HC_{i}+\beta_{4}CF_{i}+\beta_{5}PI_{i}+\beta_{6}RC_{i}+\varepsilon_{i}$$

Equation (4), $CMR\times II_{i}$ is the interaction term mentioned above, and the remaining variables are defined in Equation (1). By testing the significance of λ, the interaction effect is estimated.

## 4. Results

### 4.1. The Influence of Informal Institutions and CMR on Farmers’ Participation in RRE

This paper uses Stata 17.0 software (StataCorp LP, 4905 Lakeway Drive, College Station, TA, USA) to regress the factors affecting farmers’ participation in RRE. Before regressing the model, it is necessary to consider whether there is multicollinearity between variables. This paper uses the variance inflation factor for multicollinearity diagnosis. The results show that the highest VIF is 2.23, in line with the principle of independence, indicating that there is no serious multicollinearity between variables.

As shown in Table 4, among the informal institutions, disciplinary supervision regulations had the most significant impact on farmers’ participation in RRE governance, indicating that public opinion-based pressure, induced by warning disciplinary measures, could effectively influence farmers’ behavior. By ranking next in significance, value-oriented regulations positively affected farmers’ degree of participation at a 10% significance level, partially supporting Hypothesis 1. It can be concluded from these results that farmers’ participation in RRE is affected not only by their cost-benefit equations but also by spiritual satisfaction resulting from the effects on respect and reputation. However, the effect of the transmission internalization mechanism on farmers’ participation was not significant. In the course of the research, it was found that informal institutions are mainly based on pure text, including many professional terms that are incompatible with the local language, and their readability is not strong. For most uneducated middle-aged villagers, it is not easy to understand and spread. Furthermore, informal institutions can shape farmers’ behavior in cases where people reside close together and have tight communication networks. However, these characteristics have become less prevalent as population mobility has increased in recent years.

The influence of the CMR on farmers’ participation in RRE is significant, and the coefficient is positive, so hypothesis 2 is verified. It is evident that a strong CMR can enhance mutual recognition and mobilize farmers to participate effectively. Village cadres communicating with farmers frequently may reduce information asymmetry and encourage farmers to balance individual goals with collective ones that, in the long run, will drive farmers to actively improve their residential environment from a collective standpoint [26].

Control variables. Among household characteristics, age and public status have a significant impact on farmers’ participation. Since older farmers are more likely to be affected by their habits and convenient conditions, leading them to be less likely to respond behaviorally. Farmers with village cadres or party members at home are easy to play an exemplary role in RRE. Farmers’ cognitive status has a significant positive impact on farmers’ participation in RRE. The deeper the farmers’ understanding of the ecological and social benefits of the RRE, the more resolute the determination and attitude toward environmental improvement. The impact of policy subsidies on farmers‘ participation is significantly positive. In the external natural resources, the terrain has a significant impact on farmers’ participation.

### 4.2. Results of Quantile Regression

From the benchmark regression results, it can be concluded that the factors of informal institutions and CMR assumed in this paper have a significant impact on farmers’ participation in RRE. Nevertheless, the above results do not reflect the distribution law of each explanatory variable affecting farmer participation, prompting us to adopt a quantile regression model for use in conducting additional assessments of the impacts of informal institutions and CMR on farmers’ participation. By using the previous literature for reference [40], this paper selects five quantiles of 0.10, 0.25, 0.50, 0.75, and 0.90, and the estimation results are as follows: Model (1)~(5) are shown in Table 5.

It can be seen from Table 5, CMR has a significant impact on farmers’ participation in RRE at the 25% and 70% quantiles. Figure 2 reveals that the impact of CMR on farmers’ participation followed a “non-standard M-type” trend. Meanwhile, Table 5 results indicate that value-oriented regulations are significant at the 10% and 90% quantiles, with results presented in Figure 2 showing that farmers’ participation followed a “V-shaped” trend of initial decline, followed by a final increase. By contrast, disciplinary supervision regulations significantly influenced farmers’ participation at the 25% and 50% quantiles, thus indicating that disciplinary supervision regulation has a better governance effect on samples in the middle. The overall trend of disciplinary supervision’s influence on farmers’ participation followed an upward and then a downward trend. After an initial decreasing trend, the influence of transmission internalization on farmers’ participation rose with an increasing quantile, thus showing that the transmission internalization mechanism influenced farmers with medium and high participation the most. When taken together, these results indicate that the governance effects of informal institutions varied depending on farmers’ levels of participation in RRE governance projects.

### 4.3. Analysis of the Interactive Influence of Informal Institutions and CMR on Farmers’ Participation in RRE

Through the above analysis, it can be concluded that the impact of informal institutions on farmers’ participation in RRE is not completely significant. As the real environment of the village changed, so did the effectiveness of informal institutions. It is worth exploring whether a strong CMR can enhance the impact of informal institutions on farmers’ participation in RRE. An interaction term is added to Equation (1) to answer this question. Table 6 presents that the disciplinary supervision and CMR interaction term was significant at the 5% confidence level, thus demonstrating that a strong CMR could enhance the disciplinary supervision regulation constraint effect. This result may be explained by the fact that during the process of implementing disciplinary supervision regulations, village cadres reprimand, educate, and guide violating villagers. Those farmers who have good relationships with village cadres tend to accept their persuasion and instructions more quickly and correct their irregular behavior sooner. Meanwhile, CMR and transmission internalization mechanism interaction term were significantly positive at the 5% confidence level. This illustrates that a close and powerful CMR reasserted the influence of the transmission internalization mechanism on farmers’ participation in RRE governance since frequent CMR communication promoted learning by village cadres of the opinions of the masses. Based on the feedback from farmers, village cadres would revise the irregular and unreasonable parts of the informal institutions. Through this process, farmers move from passive imposition to heartfelt acceptance.

### 4.4. Endogeneity Problem

In order to solve the endogeneity problem, this paper uses the instrumental variable method for two-stage estimation [41]. In this paper, we use the question “Does your family belong to the big surname family in the village? Yes = 1; No = 0” as the instrumental variable of CMR. This variable is selected based on the following considerations: First, in terms of correlation, family members with large surnames have a numerical advantage. As a result, in the election of village committee members, clan members have a higher probability of obtaining voters’ votes and becoming village officials. Then, the relationship between village cadres and farmers will further deepen. Secondly, from the exogenous point of view, whether it is a large family name is difficult to affect farmers’ environmental governance behavior. In this paper, the average values of value-oriented, disciplinary supervision and transmission internalization mechanism implementation in other villages except this village are selected as their instrumental variables for the following reasons: The first is relevance. Communes will be influenced by other neighboring communes in the setting and implementing informal institutions contributes to the high correlation between instrumental variables and the implementation of informal institutions in the village. The second is exogenous. There is no justification that informal institutions in other villages affect the environmental governance behavior of farmers in their own villages.

Table 7 reports the results of the instrumental variable regression for CMR and informal institutions. Kleibergen-Paap rk LM’s *p*-value of 0.000 rejected the original hypothesis of “unidentifiable”; The Cragg-Donald Wald F value is much greater than 10, indicating that the instrumental variable selected in this article is not a weak instrumental variable. From the regression results of the first stage, it can be seen that the instrumental variables selected in this paper are effective. Furthermore, compared with the OLS estimation results, the coefficient direction and significance level of CMR in 2sls did not change significantly.

## 5. Discussion

RRE has the attribute of public goods, and environmental property rights cannot be clearly defined [42]. Farmers often have a passive attitude and a wait-and-see mentality in the process of RRE, which frequently leads to the phenomenon of the “tragedy of the commons.” It is necessary to optimize and strengthen the construction of environmental institutions and reduce free riding by farmers. In rural China, which has special social forms, institutional rules represent exogenous forces, which are difficult to embed in relationships and human society [21]. In other words, if an institution cannot be integrated into the structure of rural society, it will be high in cost and low in effectiveness [43]. The important contribution of this paper is to put the informal institutions and CMR elements into a unified framework to verify the relevant inferences with actual survey data.

There are significant regional differences in the cultural customs of rural society [44]. When promoting environmental governance, adapting national unified policies and measures to these different conditions is difficult. However, rural grass-roots organizations originated and developed according to the needs of the specific situation, and the organizing rules that they developed reflect the common will of farmers and have a greater binding force. Informal institutions play an important role in regulating farmers’ behavior in various rural affairs [45]. Therefore, this paper focuses on informal institutions, examines their impact on farmers’ participation in RRE, and confirms the conclusions of the relevant literature. For example, different informal institutions will have different impacts on farmers’ environmental governance behaviors [46]. Specifically, as shown in model (2) in Table 4, the effects of value-oriented and disciplinary supervision regulations are statistically significant. This is consistent with previous findings on environmental governance behaviors of farmers in terms of green production and concentrated disposal of municipal solid waste [22,47] and again confirms the importance of informal institutions in encouraging farmers to participate in RRE. However, the transmission internalization mechanism did not play its expected role; instead, there is a “relative institutional failure” situation. Based on the actual situation of the research area, the rural community consciousness declines with the acceleration of rural population mobility. The participation of some members in village public affairs continues to decline, and the collective action mechanism becomes loose. Although informal institutions provide an important organizational basis and behavioral norms for village environmental governance, their binding force is increasingly showing signs of weakness. Even in some villages, formalization and virtualization of informal institutions have occurred.

Village cadres are a communication bridge among farmers, which helps them reach a consensus and improve their initiative and consciousness to participate in environmental governance. It has been verified by empirical analysis that CMR is the key to improving RRE, as shown in model (2) in Table 5. CMR could lead farmers to participate in RRE in many ways. First, as a key minority group, village cadres play a leading role in rural governance. In the intimate association between cadres and the masses, the thoughts and behaviors of village cadres exert a demonstration role. Second, village cadres play a key role in connecting the government and farmers [26]. Frequent association between cadres and the masses helps in the formation of a two-way communication mechanism of RRE and ensures the accuracy of policy information conveyed. Finally, a good relationship between cadres and the masses can improve farmers’ emotional attitude towards village cadres and their trust in the institution, which can guide farmers’ initiative to participate in village public affairs.

It is noteworthy that the interaction terms between CMR and disciplinary supervision, CMR, and transmission internalization mechanisms are statistically significant. The results of this paper show that good CMR can reshape the village transmission internalization mechanism and bring its influence back into play. This is because the interaction between cadres and the masses is a process wherein the village cadres can comprehend the perspectives of the masses and communicate with them, which can provide a channel for ordinary villagers to express their wishes and actual demands. This solves the problem of unstandardized and undemocratic informal institutions in practice and also unifies the value tendency of farmers and informal institutions, which can internalize the transmission of informal institutions into the conscious activity of farmers. Furthermore, the alienation of CMR will inevitably increase the mobilization cost of RRE, thus weakening the effect of policy implementation. Therefore, effective governance of RRE cannot be achieved without the dual power of informal institutions and the CMR.

It should be noted that the possible shortcomings of this study are: First, the CMR was the core explanatory variable of this study. However, limited data permits only a single Likert5 scale to be used for quantification that, in some cases, may affect the model estimation accuracy. Warranting future development of more mature measurement scales that may enhance the scientific validity of research results. Secondly, the household survey data used in this study were limited by the questionnaire design, the time of data collection, and the study area. As a result, it is not sufficient to reflect the current situation of RRE governance in the country. In future studies, the scope should be expanded to cover a greater number of regions and a more diverse set of public affairs governance systems in order to test these findings in different contexts.

## 6. Conclusions

Based on the survey data of 1804 farmers in China, this paper, from the perspective of mobilization governance, empirically analyzes the impact of informal village institutions, the cadre-mass relationship, and their interaction on farmers’ participation in RRE governance. It finds that informal institutions promoted farmers’ participation through three mechanisms, with disciplinary supervision impact being the most significant and value-oriented regulations next in significance, but transmission internalization mechanism impact did not pass the significance test. Meanwhile, establishing a close CMR can significantly promote farmers’ participation in RRE governance, and a close CMR can reinforce the impact of disciplinary supervision and transmission internalization mechanisms on farmers’ participation. In addition, from the quantile regression model, there is a general difference in informal institutions’ governance effect on farmers with varying degrees of participation. Specifically, the impact of value-oriented regulations on farmers’ participation shows a “V-type” trend. The overall trend of the impact of disciplinary supervision on farmers’ participation is rising and then declining. The effect of the transmission internalization mechanism on farmers’ participation exhibits a “non-standard W-type” trend.

According to the conclusions obtained in this study, the following implications can be made: First, the government must design diversified informal institutions and implement them according to the actual conditions of villages. When setting up the content and form of village regulations, it should not only combine with the current background but also conform to the local customs and the actual needs of farmers. Moreover, the implementation of the informal institutions must fully respect the wishes of farmers so that they become important participants in the formulation of the institution. Second, the rural committee should encourage village cadres and farmers to move from alienation to closeness and cultivate a tight CMR. Village cadres can be deeply involved in the daily lives of villagers to seek advice, information about their needs, and comments for administration from the farmers and narrow the psychological gap between farmers and the institution. At the same time, the construction of an online and offline interaction platform between cadres and the masses can help village cadres understand public opinion faster and realize effective governance of the rural environment under the background of village labor outflow. Third, the implementation of the village institutions should pay attention to the timeliness of institutions. In the initial period when farmers’ participation in RRE is limited, it is necessary to give full play to the promoting role of various institutions. In the middle stage of RRE, the government should focus on strengthening the constraining position of disciplinary supervision regulations on farmers. Farmers’ engagement is stronger in the later stages; therefore, the government must fully utilize the benefits of value-oriented and transmission internalization mechanisms to promote farmers’ ongoing participation.

## Figures and Tables

**Figure 1 ijerph-20-00003-f001:**
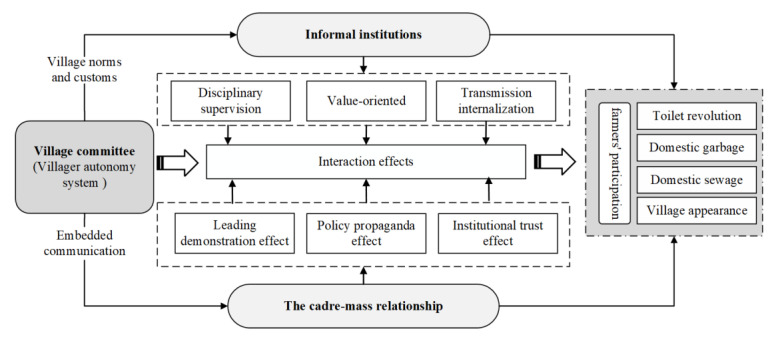
The Influence Mechanism of informal institutions and CMR on Farmers’ Participation in RRE Governance.

**Figure 2 ijerph-20-00003-f002:**
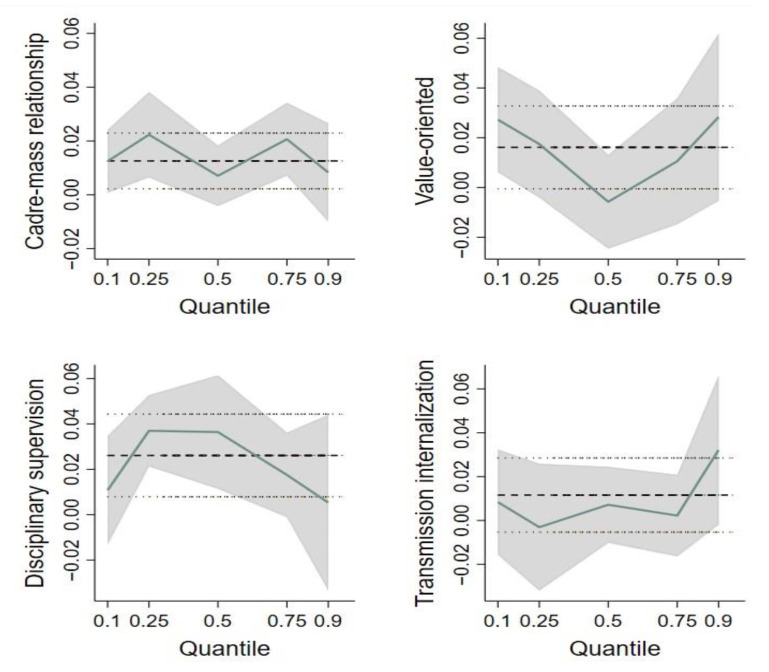
The trend of quantile regression. Note: The thick dashed line in the middle represents the OLS regression result of each variable, and the two thinner dashed horizontal lines represent confidence intervals for OLS regression results (95% confidence); The solid line represents the quantile regression results for each variable; The gray shading shows the confidence interval (95% confidence level) of the quantile regression results.

**Table 1 ijerph-20-00003-t001:** Evaluation index institutions of farmers’ participation in RRE.

Target Layer	Criteria Layer	Index Layer	Variable Weight
Toilet revolution	Household toilet type	Whether to use sanitary dry toilets or flushing toilets: In contrast to ordinary open dry toilets, sanitary dry toilets refer to closed household toilets with small manure treatment equipment that can prevent odor and pollution. Yes = 1, no = 0;	0.057
	Toilet manure treatment	Whether toilet waste is harmlessly treated: Corresponding to direct discharge, harmless manure treatment refers to farmers using their septic tanks, and cleaners to collect or through the village public sewage pool and other ways to discharge. Yes = 1, no = 0;	0.067
Domestic waste treatment	Garbage disposal method	Whether household garbage is disposed of centrally: In contrast to random disposal or incineration landfill, centralized garbage disposal refers to farmers throwing garbage into public garbage cans, garbage trucks, garbage houses, etc. Yes = 1, no = 0;	0.029
	Domestic garbage classification	Whether household garbage is treated by classification: garbage classification refers to source classification as domestic waste. Yes = 1, no = 0;	0.186
	Garbage fee	Whether the garbage disposal fee is paid to the village committee. Yes = 1, no = 0;	0.066
Domestic sewage treatment	Domestic sewage treatment method	Whether domestic sewage is treated by sewage facilities: in contrast to the direct discharge of sewage into public areas, the use of sewage treatment facilities refers to the discharge of sewage by farmers through sewage pipe networks, village public sewage treatment facilities, or self-built decentralized sewage treatment facilities. Yes = 1, no = 0;	0.115
	Sewage costs	Whether the sewage treatment costs are paid to the village committee. Yes = 1, no = 0;	0.297
Village appearance	Rural Infrastructure Maintenance Action	Whether to participate in the rural infrastructure maintenance action. Yes = 1, no = 0;	0.117
	Village greening and beautification	Whether the village is greened by planting fruits and vegetables, flowers and trees, etc. Yes = 1, no = 0;	0.046
	Village Public Environment	Outside the prescribed scope, whether there is private construction of public hidden buildings, posting advertisements, random picture walls, and other behaviors that affect the public environment of the village. Yes = 1, no = 0;	0.020

**Table 2 ijerph-20-00003-t002:** Factor analysis.

Variable		Variable Interpretation	KMO	Bartlett’s Test	Cronbach’s α
Cadre-mass relationship	interaction	Frequency of your interaction with village cadres	0.731	2337.288 (0.000)	0.850
	trust	Your level of trust in village cadres			
	support	The degree of support from village cadres			

**Table 3 ijerph-20-00003-t003:** Variable definition and descriptive statistics.

Variable		Variable Definition	Mean	S.D.
Informal institutions	Value-oriented	Whether the village has honorary recognition institutions (health model households, clean farm households, etc.). Yes = 1, no = 0	0.660	0.474
	Disciplinary supervision	Whether the village has punitive provisions for environmental damage (notice of notification, criticism, and education, etc.). Yes = 1, no = 0	0.725	0.446
	Transmission internalization	Whether agree with the relevant content of village regulations. Yes = 1, no = 0	0.545	0.498
Cadre-mass relationship	CMR	Based on factor analysis	0.000	1.000
Control variables (CV)	Age	Age of farmers (year)	61.710	11.034
	Education	Education level of farmers. not attended school = 1, primary school = 2, junior high school = 3, senior high school or technical secondary school = 4, college/junior college and above = 5	2.547	0.980
	Labor outflow	Number of family outflows/total number (%)	0.344	0.245
	Public status	Is there a village cadre or party member at home? Yes = 1, no = 0	0.121	0.326
	Income	Family per capita income (take logarithm)	9.470	0.825
	Cognition	Cognition of rural residential environment governance ^a^	2.878	1.056
	Neighborhood relations	The degree of harmony of your village neighborhood relations ^b^	2.894	1.017
	Policy subsidy	The total number of projects that can receive cash or in-kind subsidies	1.986	0.855
	Distance to town	Distance from the village to the nearest town (km)	15.341	7.090
	Terrain	Plain = 1, otherwise = 0;	0.784	0.412

^a,b^ all represents a five-point Likert scale. ^a,b^: lower layer = 1, lower middle layer = 2, middle layer = 3, upper middle layer = 4, upper layer = 5.6.

**Table 4 ijerph-20-00003-t004:** The influence of informal institution informal institutions and CMR on farmers’ participation in RRE.

Variable		Model 1	Model 2
Cadre-mass relationship	CMR		0.019 ** (0.005)
Informal institutions	Value-oriented		0.015 * (0.008)
	Disciplinary supervision		0.026 *** (0.009)
	Transmission internalization		0.05 (0.009)
Control variables	Age	−0.002 *** (0.001)	−0.002 *** (0.001)
	Education	0.006 (0.005)	0.006 (0.005)
	Labor outflow	0.020 (0.016)	0.019 (0.016)
	Public status	0.025 * (0.013)	0.023 * (0.013)
	Income	0.005 (0.005)	0.006 (0.005)
	Cognition	0.012 *** (0.004)	0.009 * (0.004)
	Neighborhood relations	0.015 *** (0.004)	0.006 (0.005)
	Policy subsidy	0.010 ** (0.003)	0.007 * (0.005)
	Distance to town	−0.001 (0.001)	−0.001 (0.001)
	Terrain	0.025 *** (0.011)	0.027 ** (0.011)
	Constant term	0.320 *** (0.055)	0.331 *** (0.057)
R-squared		0.047	0.062
Prob > F		0.000	0.000
Observations		1804	1804

*, **, *** was significant at 1%, 5% and 10% levels, respectively. standard errors are in brackets.

**Table 5 ijerph-20-00003-t005:** Results of quantile regression.

Variable	Quantile				
	0.10	0.25	0.50	0.75	0.90
Cadre-mass relationship	0.012 ** (0.006)	0.023 *** (0.008)	0.007 (0.007)	0.021 *** (0.006)	0.010 (0.011)
Value-oriented	0.026 ** (0.012)	0.017 (0.011)	−0.003 (0.010)	0.015 (0.010)	0.028 * (0.015)
Disciplinary supervision	0.010 (0.010)	0.038 *** (0.012)	0.036 *** (0.014)	0.017 (0.012)	0.005 (0.021)
Transmission internalization	0.008 (0.012)	−0.004 (−0.013)	0.007 (0.011)	0.002 (0.011)	0.032 ** (0.016)
Control variables	YES	YES	YES	YES	YES
Pseudo R2	0.051	0.033	0.021	0.043	0.093
Prob > F	0.000	0.000	0.000	0.000	0.000
Observations	1804	1804	1804	1804	1804

*, **, *** was significant at 1%, 5% and 10% levels, respectively; standard errors are in brackets.

**Table 6 ijerph-20-00003-t006:** Interaction effect test.

Variable	Model 3	Model 4	Model 5
CMR	0.013 ** (0.005)	0.013 ** (0.005)	0.012 ** (0.005)
Value-oriented	0.015 * (0.008)	0.016 * (0.008)	0.016 * (0.008)
Disciplinary supervision	0.027 *** (0.009)	0.027 *** (0.009)	0.026 *** (0.009)
Transmission internalization	0.011 (0.009)	0.012 (0.009)	0.013 (0.009)
CMR ×Value-oriented	−0.005 (0.008)		
CMR ×Disciplinary supervision		0.015 * (0.008)	
CMR ×Transmission internalization			0.014 * (0.008)
Control variables	YES	YES	YES
Pseudo R2	0.061	0.062	0.062
Pro>chi2	0.000	0.000	0.000
Observations	1804	1804	1804

*, **, *** was significant at 1%, 5% and 10% levels, respectively; standard errors are in brackets.

**Table 7 ijerph-20-00003-t007:** The instrumental variable estimation results of CMR and informal institutions.

The First Stage of 2SLS				
Variable	CMR	Value-Oriented	Disciplinary Supervision	Transmission Internalization
Big surname family	0.157 ***			
	(0.036)			
Average of value-oriented regulations (Except for the respondent’s village)		−43.688 ***	−4.481 ***	0.614 **
		(0.240)	(0.582)	(0.232)
Average of disciplinary supervision (Except for the respondent’s village)		2.153 ***	−23.857 ***	−1.196 ***
		(0.225)	(0.945)	(0.202)
Average of transmission internalization (Except for the respondent’s village)		1.551 ***	−3.626 ***	−43.208 ***
		(0.228)	(0.607)	(0.250)
Control variables	YES	YES	YES	YES
**The Second Stage of 2SLS**				
CMR	0.130 ***			
	(0.058)			
Value-oriented regulations			0.011	
			(0.009)	
Disciplinary supervision			0.050 ***	
			(0.016)	
Transmission internalization			0.015	
			(0.009)	
Control variables	YES		YES	
Kleibergen-Paap rk LM	517.557 ***		520.227 ***	
	(0.000)		(0.000)	
Cragg-Donald Wald F	19.189		448.258	
Observations	1804		1804	

**, *** was significant at 1% and 5% levels, respectively; standard errors are in brackets.

## Data Availability

The data used to support the findings of this study are available from the corresponding author upon request.

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
