# Peer review of "Effect of Village Informal Institutions and Cadre-Mass Relationship for Farmers’ Participation in Rural Residential Environment Governance in China"

_ijerph, 2022, doi:10.3390/ijerph20010003_

Round 1

Reviewer 1 Report

This paper from the perspective of mobilization governance empirically analyzes the impact of village informal institutions, the cadre-mass relationship, and their interaction on farmers’ participation in RRE governance through the entropy method, ols, and quantile regression model, which is meaningful and interesting. However, some issues have to be reconsidered.

As for the Introduction, it seems to me that the topic of this study is not very clear and is organized in a loose way. It is understandable that lines 32-74 set a background for the study but this part is too long. The key point of the Introduction should be the informal institutions and the cadre-mass relationship. Therefore, it would be better to introduce them as quickly as possible. Also, the goal and meaning of this study need to be emphasized in the Introduction.

Although the article has illustrated that village cadres perform important roles of "government agents" and "farmers’ stewards", this special system that are unique in the world should be introduced in more detail. It is helpful for readers who are not familiar with the Chinese political system.

Also, there are too many abbreviations in this article, such as IIs, CMR, DS, and TI. When they were shown in the article firstly, their original form should come together in order to avoid ambiguous understanding. Or, some abbreviations are not necessary. Maybe using these words with the complete spelling is clear and better.

Author Response

Response to Reviewer 1 Comments

Point 1: This paper from the perspective of mobilization governance empirically analyzes the impact of village informal institutions, the cadre-mass relationship, and their interaction on farmers’ participation in RRE governance through the entropy method, ols, and quantile regression model, which is meaningful and interesting. However, some issues have to be reconsidered.

As for the Introduction, it seems to me that the topic of this study is not very clear and is organized in a loose way. It is understandable that lines 32-74 set a background for the study but this part is too long. The key point of the Introduction should be the informal institutions and the cadre-mass relationship. Therefore, it would be better to introduce them as quickly as possible. Also, the goal and meaning of this study need to be emphasized in the Introduction.

Response 1: Respected reviewer thank you very much for your valuable suggestions. Your suggestions made a huge improvement in our research paper. We have tried our best to improve this manuscript as per your suggestions. Thus, we have rewritten several passages and adjusted the introduction structure of the article, deleted a large number of previously cited policy texts, and introduced the research questions and key variables of this article as soon as possible. All the changes are marked in red in the revised manuscript for my comments.The revised contents are as follows (please see lines 32 to 136)

Point 2: Although the article has illustrated that village cadres perform important roles of "government agents" and "farmers’ stewards", this special system that are unique in the world should be introduced in more detail. It is helpful for readers who are not familiar with the Chinese political system.

Response 2: Respected reviewer thank you so much for the suggestions and comments you gave us. According to your suggestion, we further introduce the villagers ' self-governance system in the manuscript.The revised contents are as follows.

Add 1:In China, villager autonomy is the fundamental political system to maintain rural governance, and committees elected by villagers are responsible for managing rural public affairs. The village committee forms a set of village regulations based on local customs and farmers' consensus opinions. These regulations are the important institutional basis of villager autonomy, which enables members to carry out self-management, self-education, and self-restraint. Village regulation is the most significant and extensive part of the Chinese rural informal institutions. Village regulation is the most significant and extensive part of the Chinese rural informal institutions. Thus, what is the effect of the informal institutions represented by village regulations on RRE? The question deserves further consideration.

Add 2:As far as RRE governance is concerned, the positive interactive relationship between village cadres and villagers is embodied in the exchange and sharing of resources for the unified goal. Eventually, they form a long-term and stable relationship of mutual dependence and cooperation. The quality of the relationship between cadres and farmers will further affect the mobilization effect of farmers. Studies indicate that the close cadre-mass relationship impacted agricultural waste recycling, infrastructure management, and land expropriation compensation efforts, and maintains village discipline while standardizing farmers' cooperation and collective action.

All the changes are marked in red in the revised manuscript for my comments.

 Point 3: Also, there are too many abbreviations in this article, such as IIs, CMR, DS, and TI. When they were shown in the article firstly, their original form should come together in order to avoid ambiguous understanding. Or, some abbreviations are not necessary. Maybe using these words with the complete spelling is clear and better.

Response 3: Dear reviewer thank you again for your valuable suggestions and comments. We use the full spelling of IIs, CMR, DS, and TI to replace abbreviations. We use the full spelling of informal institutions, value-oriented, disciplinary supervision, and transmission internalization to replace abbreviations IIs, VO, DS, and TI. However, due to the long spelling of the word group cadre-mass relationship, rural residential environment, it is hoped that the abbreviations of these two phrases will be retained.

Reviewer 2 Report

1. Lot of theoretical and experimental data are presented in the paper. Since it is a research article (not a review) suggesting to present only important findings in an attractive way.

What are the usefulnesses of your findings to general public ?

Make a separate topic for conclusion. It should not come with the discussion. 

Author Response

Point 1: Lot of theoretical and experimental data are presented in the paper. Since it is a research article (not a review) suggesting to present only important findings in an attractive way.

Response 1: Respected reviewer thank you very much for your valuable suggestions. Your suggestions made a huge improvement in our research paper. As an empirical research article, we are apologetic for confusing you. Thus, we have rewritten several passages and are confident that the revised manuscript is much improved with regard to this critical point. Specifically, we adjusted the introduction structure of the article, deleted a large number of previously cited policy texts, and introduced the research questions and key variables of this article as soon as possible. All the changes are marked in red in the revised manuscript for my comments.

Point 2: What are the usefulnesses of your findings to general public?

Response 2: Dear reviewer thank you again for your valuable suggestions and comments. The fundamental starting point of the renovation of rural living environment is to improve the well-being of people 's livelihood, so that farmers have a good production and living environment in the renovation process, and improve the effect of modern rural construction. The usefulness of manuscript discovery to the general public is as follows: Firstly, in terms of the core variable in the manuscript, cadre-mass relationship. A good relationship between cadres and mass guarantees the democratic rights of farmers to participate in village affairs. Through frequent communication with the villagers, village cadres go deep into the daily lives of the villagers, solicit opinions from the farmers, understand their needs, as well as opinions on administrative management, which helps to narrow the psychological gap with the farmers. Secondly, due to the geographical dispersion of rural China and the complexity of its social environment as well as the heterogeneity of its historical and cultural foundations, the effect of government environmental regulation is limited. rural China has the characteristics of “acquaintance society”, and the influence of informal forces such as customs, social networks and human relations on farmers cannot be ignored. Through specific measures such as honorary titles, criticism and education, environmental village regulations can greatly satisfy farmers’ demands for honor, reputation, and guide farmers’ environmental behaviors. Meanwhile, it also plays an invisible regulatory role for farmers, shaping their collective consciousness and the overall rural concept. Finally, the public’s deep participation in RRE governance is of great practical significance to comprehensively improve the quality of the rural living environment, narrow the gap between urban and rural areas and build a moderately prosperous society in an all-around way.

 Point 3: Make a separate topic for conclusion. It should not come with the discussion.

Response 3: Thank you for the constructive comments on my manuscript. According to the suggestion, we make a separate topic for the conclusion and reword the inappropriate part of the conclusion. All the changes are marked in red in the revised manuscript for my comments.

Reviewer 3 Report

Comments and Suggestions for Authors

ijerph-2013223

The article presents a valuation of multiple aspects of the Effect of village informal Institutions and cadre mass relationship for farmers’ participation in rural residential environment governance in China

This article is well written, well structured, and uses an extended and up-to-date set of references. This post also provides interesting background information on the problem described and a few minor issues that came up during my review:

1. The English grammar and style should be checked throughout the paper.

2. Author needs to add clear objectives for the study.     

3. What does your article bring to the research field that other papers did not addressed. I think this must be clearly established to highlight the reader about the novelty statement of this article.

4. The figure one can be improved in:

1.         Quality and resolution

2.         Make sure they are created by the authors ( original) or at least properly cited.

3.         Make sure that figure 2 is created by the authors ( original) or at least properly cited

5. Pay more attention to referencing Tables properly

6. Conclusions and recommendations must be clearly related to the results. These relationships should be included in the text.

7. The authors should mention the main limitations of this study at the end of the conclusion section in one paragraph.

Author Response

Point 1: The article presents a valuation of multiple aspects of the Effect of village informal Institutions and cadre mass relationship for farmers’ participation in rural residential environment governance in China. This article is well written, well structured, and uses an extended and up-to-date set of references. This post also provides interesting background information on the problem described and a few minor issues that came up during my review: The English grammar and style should be checked throughout the paper

Response 1: Respected reviewer thank you very much for your valuable suggestions. Your suggestions made a huge improvement in our research paper. According to your Point 1, We have double-checked mistakes and have polished this manuscript by AJE.

Point2: Author needs to add clear objectives for the study.

Response 2: Dear reviewer after your valuable advice we have added clear objectives for the study as follows:

Based on the survey data of 1804 farmers in Shaanxi Province, China, this paper from the perspective of mobilization governance, empirically analyzes the impact of village informal institutions, the cadre-mass relationship, and their interaction on farmers' participation in RRE governance through the entropy method, ols and quantile regression model. Specifically, this research addresses the following three main targets: (1) How, and to what extent, do informal institutions and cadre-mass relationships affect the degree of farmers' participation in RRE governance? (2) What is the interactive influence of informal institutions and cadre-mass rrelationshipson farmers' participation in RRE governance? (3) What are the changing trends in the impact of informal institutions and cadre-mass relationships on farmers' participation in RRE governance?

Point 3: What does your article bring to the research field that other papers did not addressed? I think this must be clearly established to highlight the reader about the novelty statement of this article.

Response3: Respected reviewer thank you again for your comments. To the best of our knowledge, the contribution of this paper to the existing literature is as follows. On the one hand, concerning the research subject, as China's RRE governance advances, differences in farmers' behavior are no longer simply a matter of whether or not to participate, but more in the degree of participation. However, existing research on the degree of farmers' participation has mostly focused on a particular aspect of RRE governance or is based on farmers' evaluation of their own participation. There is a lack of multidimensional measurement of farmers' actual participation in RRE governance. Thus, this study explores and consummates the evaluation index system to measure the degree of farmers' participation in RRE governance, and enrichment of the existing literature. On the other hand, concerning the research perspective, under the villager autonomy system of rural society in China, environmental village regulations and village cadres are important factors in mobilizing farmers to participate in the governance of public affairs. Institutions are undifferentiated for any individual. However, in the "pattern of difference sequence " centered on village cadres, village cadres cannot be undifferentiated for all villagers. What is the impact of the cadre-mass relationship on the implementation of the village informal institutions, and whether there is an interaction between the two is worth exploring. However, most of the previous studies analyzed the rural informal institutions or CMR from a single perspective, rarely both into a unified framework for specialized exploration.

Point 4: Figure one can be improved in:

  1. Quality and resolution
  2. Make sure they are created by the authors (original) or at least properly cited.
  3. Make sure that figure 2 is created by the authors (original) or at least properly cited

Response 4: We sincerely appreciate the reviewer's comments. We complied with the requests of the reviewer(s). Figure 1 and Figure 2 are improved in the revised manuscript. As far as Figure 1 is concerned, we re-compose it with the content of the article and the modification suggestions. As far as Figure 2 is concerned, we plotted figure 2 on the basis of quantile regression using Stata17.0, to more directly reflect the differences in the impact of informal institution and cadre-mass relationship on the degree of farmers' participation in different quantiles. Further upon the request of the reviewer, we have now improved the resolution of images and presented better quality images in the revised Figure 2.

Point 5: Pay more attention to referencing Tables properly

Response 5: We are thankful to the reviewer for this suggestion. We have checked the way of citing tables in the manuscript one by one, and corrected the unreasonable places. The modified contents were marked with red pen in lines 251, 256 and 375.

 Point 6: Conclusions and recommendations must be clearly related to the results. These relationships should be included in the text.

Response 6: We appreciate the reviewer’s advice and we tried to be more cautious in style and conclusions. We correspond the conclusions and recommendations to the results.

Result1: The influence of informal institutions and CMR on farmers’ participation in RRE.

Conclusion1: It finds that informal institutions promoted farmers’ participation through three mechanisms, with disciplinary supervision impact being the most significant and value-oriented regulations next in significance, but transmission internalization mechanism impact did not pass the significance test.

Recommendation 1: The government must design diversified informal institutions and implement them according to the actual conditions of villages. When setting up the content and form of village regulations, it should not only combine with the current background but also conform to the local customs and the actual needs of farmers. Moreover, the implementation of the informal institutions must fully respect the wishes of farmers, so that they become important participants in the formulation of the institution.

Result 2: Analysis of the interactive influence of informal institutions and CMR on farmers’ participation in RRE

Conclusion2: Establishing a close CMR can significantly promote farmers' participation in RRE governance, and a close CMR can reinforce the impact of disciplinary supervision and transmission internalization mechanisms on farmers' participation.

Recommendation 2: The rural committee should encourage village cadres and farmers to move from alienation to closeness and cultivate a tight CMR. Village cadres can be deeply involved in the daily lives of villagers to seek advice, information about their needs, and com-ments for administration from the farmers and narrow the psychological gap between farmers and the institution. At the same time, the construction of an online and offline interaction platform between cadres and the masses can help village cadres understand public opinion faster and realize effective governance of the rural environment under the background of village labor outflow.

Result3: Results of Quantile Regression

Conclusion3: In addition, from the quantile regression model, there is a general difference in informal institutions' governance effect on farmers with varying degrees of participation. Specifically, the impact of value-oriented regulations on farmers' participation shows a " V-type" trend. The overall trend of the impact of disciplinary supervision on farmers' participation is rising and then declining. The effect of the transmission internalization mechanism on farmers’ participation exhibits a “non-standard W-type” trend.

Recommendation 3: The implementation of the village institutions should pay attention to the timeliness of institutions. In the initial period when farmers’ participation in RRE is limited, it is necessary to give full play to the promoting role of various institutions. In the middle stage of RRE, the government should focus on strengthening the constraining position of disciplinary supervision regulations on farmers. Farmers’ engagement is stronger in the later stages; therefore, the government must fully utilize the benefits of value-oriented and transmission internalization mechanism to promote farmers’ ongoing participation. (Or please see lines 493 to 530)

Point 7: The authors should mention the main limitations of this study at the end of the conclusion section in one paragraph.

Response 7: Dear reviewer thank you again for your valuable suggestions and comments. The main limitations of this study are modified as follows(Or please see lines 482 to 493).

It should be noted that the possible shortcomings of this study are: First, the CMR was the core explanatory variable of this study. However, limited data permits only a single Likert5 scale to be used for quantification that, in some cases, may affect the model estimation accuracy. Warranting future development of more mature measurement scales that may enhance the scientific validity of research results. Secondly, the household survey data used in this study were limited by the questionnaire design, the time of data collection and the study area. As a result,it is not sufficient to reflect the current situation of RRE governance in the country. In future studies, the scope should be expanded to cover a greater number of regions and a more diverse set of public affairs governance systems in order to test these findings in different contexts.

Round 2

Reviewer 1 Report

The authors have made relatively perfect modifications according to the review comments, and it is recommended to accept.